# LOW-RANK TRAINING OF DEEP NEURAL NETWORKS FOR EMERGING MEMORY TECHNOLOGY

## ABSTRACT

The recent success of neural networks for solving difficult decision tasks has incentivized incorporating smart decision making "at the edge." However, this work has traditionally focused on neural network *inference*, rather than *training*, due to memory and compute limitations, especially in emerging non-volatile memory systems, where writes are energetically costly and reduce lifespan. Yet, the ability to train at the edge is becoming increasingly important as it enables real-time adaptability to device drift and environmental variation, user customization, and federated learning across devices. In this work, we address two key challenges for training on edge devices with non-volatile memory: low write density and low auxiliary memory. We present a low-rank training scheme that addresses these challenges while maintaining computational efficiency. We then demonstrate the technique on a representative convolutional neural network across several adaptation problems, where it out-performs standard SGD both in accuracy and in number of weight writes.

## 1 INTRODUCTION

Deep neural networks have shown remarkable performance on a variety of challenging inference tasks. As the energy efficiency of deep-learning inference accelerators improves, some models are now being deployed directly to edge devices to take advantage of increased privacy, reduced network bandwidth, and lower inference latency. Despite edge deployment, training happens predominately in the cloud. This limits the privacy advantages of running models on-device and results in static models that do not adapt to evolving data distributions in the field.

Efforts aimed at on-device training address some of these challenges. Federated learning aims to keep data on-device by training models in a distributed fashion (Konecný et al., 2016). On-device model customization has been achieved by techniques such as weight-imprinting (Qi et al., 2018), or by retraining limited sets of layers. On-chip training has also been demonstrated for handling hardware imperfections (Zhang et al., 2017; Gonugondla et al., 2018). Despite this progress with small models, on-chip training of larger models is bottlenecked by the limited memory size and compute horsepower of edge processors.

Emerging non-volatile (NVM) memories such as resistive random access memory (RRAM) have shown great promise for energy and area-efficient inference (Yu, 2018). However, on-chip training requires a large number of writes to the memory, and RRAM writes cost significantly more energy than reads (e.g., 10.9 pJ/bit versus 1.76 pJ/bit (Wu et al., 2019)). Additionally, RRAM endurance is on the order of $10^6$ writes (Grossi et al., 2019), shortening the lifetime of a device due to memory writes for on-chip training.

In this paper, we present an online training scheme amenable to NVM memories to enable next generation edge devices. Our contributions are (1) an algorithm called Streaming Kronecker Sum Approximation (SKS), and its analysis, which addresses the two key challenges of low write density and low auxiliary memory; (2) two techniques "gradient max-norm" and "streaming batch norm" to help training specifically in the online setting; (3) a suite of adaptation experiments to demonstrate the advantages of our approach.

## 2 RELATED WORK

**Efficient training for resistive arrays.** Several works have aimed at improving the efficiency of training algorithms on resistive arrays. Of the three weight-computations required in training (forward, backprop, and weight update), weight updates are the hardest to parallelize using the array structure. Stochastic weight updates (Gokmen & Vlasov, 2016) allow programming of all cells in a crossbar at once, as opposed to row/column-wise updating. Online Manhattan rule updating (Zamanidoost et al., 2015) can also be used to update all the weights at once. Several works have proposed new memory structures to improve the efficiency of training (Soudry et al., 2015; Ambrogio et al., 2018). The number of writes has also been quantified in the context of chip-in-the-loop training (Yu et al., 2016).

**Distributed gradient descent.** Distributed training in the data center is another problem that suffers from expensive weight updates. Here, the model is replicated onto many compute nodes and in each training iteration, the mini-batch is split across the nodes to compute gradients. The distributed gradients are then accumulated on a central node that computes the updated weights and broadcasts them. These systems can be limited by communication bandwidth, and compressed gradient techniques (Aji & Heafield, 2017) have therefore been developed. In Lin et al. (2017), the gradients are accumulated over multiple training iterations on each compute node and only gradients that exceed a threshold are communicated back to the central node. In the context of on-chip training with NVM, this method helps reduce the number of weight updates. However, the gradient accumulator requires as much memory as the weights themselves, which negates the density benefits of NVM.

**Low-Rank Training.** Our work draws heavily from previous low-rank training schemes that have largely been developed for use in recurrent neural networks to uncouple the training memory requirements from the number of time steps inherent to the standard truncated backpropagation through time (TBPTT) training algorithm. Algorithms developed since then to address the memory problem include Real-Time Recurrent Learning (RTRL) (Williams & Zipser, 1989), Unbiased Online Recurrent Optimization (UORO) (Tallec & Ollivier, 2017), Kronecker Factored RTRL (KF-RTRL) (Mujika et al., 2018), and Optimal Kronecker Sums (OK) (Benzing et al., 2019). These latter few techniques rely on the weight gradients in a weight-vector product looking like a sum of outer products (i.e., Kronecker sums) of input vectors with backpropagated errors. Instead of storing a growing number of these sums, they can be approximated with a low-rank representation involving fewer sums.

## 3 TRAINING NON-VOLATILE MEMORY

The meat of most deep learning systems are many **w**eight matrix - **a**ctivation vector products $\boldsymbol{W} \cdot \boldsymbol{a}$. Fully-connected (dense) layers use them explicitly: $\boldsymbol{a}^{[\ell]} = \sigma \left( \boldsymbol{W}^{[\ell]} \boldsymbol{a}^{[\ell-1]} + \boldsymbol{b}^{[\ell]} \right)$ for layer $\ell$, where $\sigma$ is a non-linear activation function (more details are discussed in detail in Appendix C.1). Recurrent neural networks use one or many matrix-vector products per recurrent cell. Convolutional layers can also be interpreted in terms of matrix-vector products by unrolling the input feature map into strided convolution-kernel-size slices. Then, each matrix-vector product takes one such input slice and maps it to all channels of the corresponding output pixel (more details are discussed in Appendix C.2).

The ubiquity of matrix-vector products allows us to adapt the techniques discussed in "Low-Rank Training" of Section 2 to other network architectures. Instead of reducing the memory across time steps, we can reduce the memory across training samples in the case of a traditional feedforward neural network. However, in traditional training (e.g., on a GPU), this technique does not confer advantages. Traditional training platforms often have ample memory to store a batch of activations and backpropagated gradients, and the weight updates $\Delta \boldsymbol{W}$ can be applied directly to the weights $\boldsymbol{W}$ once they are computed, allowing temporary activation memory to be deleted. The benefits of low-rank training only become apparent when looking at the challenges of proposed NVM devices:

**Low write density (LWD).** In NVM, writing to weights at every sample is costly in energy, time, and endurance. These concerns are exacerbated in multilevel cells, which require several steps of an iterative write-verify cycle to program the desired level. We therefore want to minimize the number of writes to NVM.

**Low auxiliary memory (LAM).** NVM is the densest form of memory. In 40nm technology, RRAM 1T-1R bitcells @ 0.085 um$^2$ (Chou et al., 2018) are 2.8x smaller than 6T SRAM cells @ 0.242 um$^2$ (TSMC, 2019). Therefore, NVM should be used to store the memory-intensive weights. By the same

token, no other on-chip memory should come close to the size of the on-chip NVM. In particular, if our $b-$bit NVM stores a weight matrix of size $n_o \times n_i$, we should use at most $r(n_i + n_o)b$ auxiliary non-NVM memory, where $r$ is a small constant. Despite these space limitations, the reason we might opt to use auxiliary (large, high endurance, low energy) memory is because there are places where writes are frequent, violating LWD if we were to use NVM.

In the traditional minibatch SGD setting with batch size $B$, an upper limit on the write density per cell per sample is easily seen: $1/B$. However, to store such a batch of updates without intermediate writes to NVM would require auxiliary memory proportional to $B$. Therefore, a trade-off becomes apparent. If $B$ is reduced, LAM is satisfied at the cost of LWD. If $B$ is raised, LWD is satisfied at the cost of LAM. Using low-rank training techniques, the auxiliary memory requirements are decoupled from the batch size, allowing us to increase $B$ while satisfying both LWD and LAM[1]. Additionally, because the low-rank representation uses so little memory, a larger bitwidth can be used, potentially allowing for gradient accumulation in a way that is not possible with low bitwidth NVM weights. In the next section, we elaborate on the low-rank training method.

## 4 LOW-RANK TRAINING METHOD

Let $\boldsymbol{z}^{(i)} = \boldsymbol{W}\boldsymbol{a}^{(i)} + \boldsymbol{b}$ be the standard affine transformation building block of some larger network, e.g., $\boldsymbol{y}_p^{(i)} = f_{post}(\boldsymbol{z}^{(i)})$ and $\boldsymbol{a}^{(i)} = f_{pre}(\boldsymbol{x}^{(i)})$ with prediction loss $\mathcal{L}(\boldsymbol{y}_p^{(i)}, \boldsymbol{y}_t^{(i)})$, where $(\boldsymbol{x}^{(i)}, \boldsymbol{y}_t^{(i)})$ is the $i^{\text{th}}$ training sample pair. Then weight gradient $\nabla_{\boldsymbol{W}}\mathcal{L}^{(i)} = \boldsymbol{dz}^{(i)}\left(\boldsymbol{a}^{(i)}\right)^\top = \boldsymbol{dz}^{(i)} \otimes \boldsymbol{a}^{(i)}$ where $\boldsymbol{dz}^{(i)} = \nabla_{\boldsymbol{z}^{(i)}}\mathcal{L}^{(i)}$. A minibatch SGD weight update accumulates this gradient over $B$ samples: $\Delta\boldsymbol{W} = -\eta \sum_{i=1}^{B} \boldsymbol{dz}^{(i)} \otimes \boldsymbol{a}^{(i)}$ for learning rate $\eta$.

For a rank-$r$ training scheme, approximate the sum $\sum_{i=1}^{B} \boldsymbol{dz}^{(i)} \otimes \boldsymbol{a}^{(i)}$ by iteratively updating two rank-$r$ matrices $\tilde{\boldsymbol{L}} \in \mathbb{R}^{n_o \times r}, \tilde{\boldsymbol{R}} \in \mathbb{R}^{n_i \times r}$ with each new outer product: $\tilde{\boldsymbol{L}}\tilde{\boldsymbol{R}}^\top \leftarrow rankReduce(\tilde{\boldsymbol{L}}\tilde{\boldsymbol{R}}^\top + \boldsymbol{dz}^{(i)} \otimes \boldsymbol{a}^{(i)})$. Therefore, at each sample, we convert the rank-$q = r + 1$ system $\tilde{\boldsymbol{L}}\tilde{\boldsymbol{R}}^\top + \boldsymbol{dz}^{(i)} \otimes \boldsymbol{a}^{(i)}$ into the rank-$r$ $\tilde{\boldsymbol{L}}\tilde{\boldsymbol{R}}^\top$. In the next sections, we discuss how to compute $rankReduce$.

### 4.1 OPTIMAL KRONECKER SUM APPROXIMATION (OK)

One option for $rankReduce(\boldsymbol{X})$ to convert from rank $q = r + 1$ $\boldsymbol{X}$ to rank $r$ is a minimum error estimator, which is implemented by selecting the top $r$ components of a singular value decomposition (SVD) of $\boldsymbol{X}$. However, a naïve implementation is computationally infeasible and biased: $\mathbb{E}[rankReduce(\boldsymbol{X})] \neq \boldsymbol{X}$. Benzing et al. (2019) solves these problems by proposing a minimum variance unbiased estimator for $rankReduce$, which they call the OK algorithm[2].

The OK algorithm can be understood in two key steps: first, an efficient method of computing the SVD of a Kronecker sum; second, a method of splitting the singular value matrix $\boldsymbol{\Sigma}$ into two rank-$r$ matrices whose outer product is a minimum-variance, unbiased estimate of $\boldsymbol{\Sigma}$. Details can be found in their paper, however we include a high-level explanation in Sections 4.1.1 and 4.1.2 to aid our discussions. Note that our variable notation differs from Benzing et al. (2019).

#### 4.1.1 EFFICIENT SVD OF KRONECKER SUMS

Let $\boldsymbol{L} = [\tilde{\boldsymbol{L}}, \boldsymbol{dz}^{(i)}]$ and $\boldsymbol{R} = [\tilde{\boldsymbol{R}}, \boldsymbol{a}^{(i)}]$ so that $\boldsymbol{L}\boldsymbol{R}^\top = \tilde{\boldsymbol{L}}\tilde{\boldsymbol{R}}^\top + \boldsymbol{dz}^{(i)} \otimes \boldsymbol{a}^{(i)}$. Recall that $rankReduce$ should turn rank-$q$ $\boldsymbol{L}\boldsymbol{R}^\top$ into an updated rank-$r$ $\tilde{\boldsymbol{L}}\tilde{\boldsymbol{R}}^\top$.

QR-factorize $\boldsymbol{L} = \boldsymbol{Q}_L\boldsymbol{R}_L$ and $\boldsymbol{R} = \boldsymbol{Q}_R\boldsymbol{R}_R$ where $\boldsymbol{Q}_L \in \mathbb{R}^{n_o \times q}, \boldsymbol{Q}_R \in \mathbb{R}^{n_i \times q}$ are orthogonal so that $\boldsymbol{L}\boldsymbol{R}^\top = \boldsymbol{Q}_L(\boldsymbol{R}_L\boldsymbol{R}_R^\top)\boldsymbol{Q}_R^\top$. Let $\boldsymbol{C} = \boldsymbol{R}_L\boldsymbol{R}_R^\top \in \mathbb{R}^{q \times q}$. Then we can find the SVD of $\boldsymbol{C} = \boldsymbol{U}_C\boldsymbol{\Sigma}\boldsymbol{V}_C^\top$ in $\mathcal{O}(q^3)$ time (Cline & Dhillon, 2006), making it computationally feasible on small devices. Now we have:

---

[1]This can alternately be achieved by sub-sampling the training data by $r/B$ where $r$ is the OK rank. The purpose of using a low-rank estimate is that for the same memory cost, it is significantly more informational than the sub-sampled data, allowing for faster training convergence.

[2]Their target application differs slightly in that they handle matrix - vector Kronecker sums rather than vector - vector Kronecker sums.

$$\boldsymbol{L}\boldsymbol{R}^{\top} = \boldsymbol{Q}_L(\boldsymbol{U}_C\boldsymbol{\Sigma}\boldsymbol{V}_C^{\top})\boldsymbol{Q}_R^{\top} = (\boldsymbol{Q}_L\boldsymbol{U}_C)\boldsymbol{\Sigma}(\boldsymbol{Q}_R\boldsymbol{V}_C)^{\top} \qquad (1)$$

which gives the SVD of $\boldsymbol{L}\boldsymbol{R}^{\top}$ since $\boldsymbol{Q}_L\boldsymbol{U}_C$ and $\boldsymbol{Q}_R\boldsymbol{V}_C$ are orthogonal and $\boldsymbol{\Sigma}$ is diagonal. This SVD computation has a time complexity of $\mathcal{O}((n_i+n_o+q)q^2)$ and a space complexity of $\mathcal{O}((n_i+n_o+q)q)$.

### 4.1.2 MINIMUM VARIANCE, UNBIASED ESTIMATE OF $\boldsymbol{\Sigma}$

In Benzing et al. (2019), it is shown that the problem of finding a rank-$r$ minimum variance unbiased estimator of $\boldsymbol{L}\boldsymbol{R}^{\top}$ can be reduced to the problem of finding a rank-$r$ minimum variance unbiased estimator of $\boldsymbol{\Sigma}$ and plugging it in to (1).

Further, it is shown that such an optimal approximator for $\boldsymbol{\Sigma} = \text{diag}(\sigma_1, \sigma_2, \ldots, \sigma_q)$, where $\sigma_1 \geq \sigma_2 \geq \cdots \geq \sigma_q$ will involve keeping the $m-1$ largest singular values and mixing the smaller singular values $\sigma_m, \ldots, \sigma_q$ within their $(k+1) \times (k+1)$ submatrix with $m, k$ defined below. Let:

$$m = \min \ i \ \text{ s.t. } \ (q-i)\sigma_i \leq \sum_{j=i}^{q}\sigma_j \qquad\qquad k = q - m$$

$$\boldsymbol{x}_0 = \left(\sqrt{1 - \frac{\sigma_m k}{s_1}}, \ldots, \sqrt{1 - \frac{\sigma_q k}{s_1}}\right)^{\top} \qquad\qquad s_1 = \sum_{i=m}^{q}\sigma_i$$

Note that $||\boldsymbol{x}_0||_2 = 1$. Let $\boldsymbol{X} \in \mathbb{R}^{(k+1)\times(k)}$ be orthogonal such that its left nullspace is the span of $\boldsymbol{x}_0$. Then $\boldsymbol{X}\boldsymbol{X}^{\top} = I - \boldsymbol{x}_0\boldsymbol{x}_0^{\top}$. Now, let $\mathbf{s} \in \{-1, 1\}^{(k+1)\times 1}$ be uniform random signs and define:

$$\boldsymbol{X}_s = (\mathbf{s} \odot \boldsymbol{X}_{:,1}, \ldots, \mathbf{s} \odot \boldsymbol{X}_{:,k}) \qquad\qquad \boldsymbol{Z} = \sqrt{\frac{s_1}{k}} \cdot \boldsymbol{X}_s$$

$$\tilde{\boldsymbol{\Sigma}}_L = \tilde{\boldsymbol{\Sigma}}_R = \text{diag}\left(\sqrt{\sigma_1}, \ldots, \sqrt{\sigma_{m-1}}, \boldsymbol{Z}\right) \qquad (2)$$

where $\odot$ is an element-wise product. Then $\tilde{\boldsymbol{\Sigma}}_L\tilde{\boldsymbol{\Sigma}}_R^{\top} = \tilde{\boldsymbol{\Sigma}}$ is a minimum variance, unbiased[3] rank-$r$ approximation of $\boldsymbol{\Sigma}$. Plugging $\tilde{\boldsymbol{\Sigma}}$ into (1),

$$\boldsymbol{L}\boldsymbol{R}^{\top} = (\boldsymbol{Q}_L\boldsymbol{U}_C)\boldsymbol{\Sigma}(\boldsymbol{Q}_R\boldsymbol{V}_C)^{\top} \approx (\boldsymbol{Q}_L\boldsymbol{U}_C)\tilde{\boldsymbol{\Sigma}}(\boldsymbol{Q}_R\boldsymbol{V}_C)^{\top} = (\boldsymbol{Q}_L\boldsymbol{U}_C\tilde{\boldsymbol{\Sigma}}_L)(\boldsymbol{Q}_R\boldsymbol{V}_C\tilde{\boldsymbol{\Sigma}}_R)^{\top} \quad (3)$$

Thus, $\tilde{\boldsymbol{L}} = \boldsymbol{Q}_L\boldsymbol{U}_C\tilde{\boldsymbol{\Sigma}}_L \in \mathbb{R}^{n_o \times r}$ and $\tilde{\boldsymbol{R}} = \boldsymbol{Q}_R\boldsymbol{V}_C\tilde{\boldsymbol{\Sigma}}_R \in \mathbb{R}^{n_i \times r}$ gives us a minimum variance, unbiased, rank-$r$ approximation $\tilde{\boldsymbol{L}}\tilde{\boldsymbol{R}}^{\top}$.

## 4.2 STREAMING KRONECKER SUM APPROXIMATION (SKS)

Although the standalone OK algorithm presented by Benzing et al. (2019) has good asymptotic computational complexity, our vector-vector outer product sum use case permits further optimizations. In this section we present these optimizations, and we refer readers to the explicit implementation called Streaming Kronecker Sum Approximation (SKS) in Algorithm 1 of Appendix A.

### 4.2.1 MAINTAIN ORTHOGONAL $\boldsymbol{Q}_L, \boldsymbol{Q}_R$

The main optimization is a method of avoiding recomputing the QR factorization of $\boldsymbol{L}$ and $\boldsymbol{R}$ at every step. Instead, we keep track of orthogonal matrices $\boldsymbol{Q}_L, \boldsymbol{Q}_R$, and weightings $\boldsymbol{c}_x$ such that $\tilde{\boldsymbol{L}} = \boldsymbol{Q}_L \cdot \text{diag}(\sqrt{\boldsymbol{c}_x})_{[:r]}$ and $\tilde{\boldsymbol{R}} = \boldsymbol{Q}_R \cdot \text{diag}(\sqrt{\boldsymbol{c}_x})_{[:r]}$. Upon receiving a new sample, a single inner loop of the numerically-stable modified Gram-Schmidt (MGS) algorithm (Björck, 1967) can be used to update $\boldsymbol{Q}_L$ and $\boldsymbol{Q}_R$. The orthogonal basis coefficients $\boldsymbol{c}_L = \boldsymbol{Q}_L^{\top}\boldsymbol{dz}^{(i)}$ and $\boldsymbol{c}_R = \boldsymbol{Q}_R^{\top}\boldsymbol{a}^{(i)}$ computed during MGS can be used to find the new value of $\boldsymbol{C} = \boldsymbol{c}_L\boldsymbol{c}_R^{\top} + \text{diag}(\boldsymbol{c}_x)$.

---

[3]The fact that it is unbiased: $\mathbb{E}[\tilde{\boldsymbol{\Sigma}}] = \boldsymbol{\Sigma}$ can be easily verified.

After computing $\tilde{\boldsymbol{\Sigma}}_L = \tilde{\boldsymbol{\Sigma}}_R$ in (2), we can orthogonalize these matrices into $\tilde{\boldsymbol{\Sigma}}_L = \tilde{\boldsymbol{\Sigma}}_R = \boldsymbol{Q}_x \boldsymbol{R}_x$. Then from (3), we have $\tilde{\boldsymbol{L}}\tilde{\boldsymbol{R}}^\top = (\boldsymbol{Q}_L \boldsymbol{U}_C \boldsymbol{Q}_x)(\boldsymbol{R}_x \boldsymbol{R}_x^\top)(\boldsymbol{Q}_R \boldsymbol{V}_C \boldsymbol{Q}_x)^\top$. With this formulation, we can maintain orthogonality in $\boldsymbol{Q}_L, \boldsymbol{Q}_R$ by setting:

$$\boldsymbol{Q}_L \leftarrow \boldsymbol{Q}_L \boldsymbol{U}_C \boldsymbol{Q}_x \qquad \boldsymbol{Q}_R \leftarrow \boldsymbol{Q}_R \boldsymbol{V}_C \boldsymbol{Q}_x \qquad \boldsymbol{c}_x \leftarrow \operatorname{diag}(\boldsymbol{R}_x \boldsymbol{R}_x^\top)$$

These matrix multiplies require $\mathcal{O}((n_i + n_o)q^2)$ multiplications, so this optimization does not improve asymptotic complexity bounds. This optimization may nonetheless be practically significant since matrix multiplies are easy to parallelize and would typically not be the bottleneck of the computation compared to Gram-Schmidt. The next section discusses how to orthogonalize $\tilde{\boldsymbol{\Sigma}}_L$ efficiently and why $(\boldsymbol{R}_x \boldsymbol{R}_x^\top)$ is diagonal.

### 4.2.2 ORTHOGONALIZATION OF $\tilde{\boldsymbol{\Sigma}}_L$

Orthogonalization of $\tilde{\boldsymbol{\Sigma}}_L$ is relatively straightforward. From (2), the columns of $\tilde{\boldsymbol{\Sigma}}_L$ are orthogonal since $\boldsymbol{Z}$ is orthogonal. However, they do not have unit norm. We can therefore pull out the norm into a separate diagonal matrix $\boldsymbol{R}_x$ with diagonal elements $\sqrt{\boldsymbol{c}_x}$:

$$\boldsymbol{Q}_x = \begin{bmatrix} \boldsymbol{I}_{m-1} & 0 \\ 0 & \boldsymbol{X}_s \end{bmatrix} \qquad \sqrt{\boldsymbol{c}_x} = (\sqrt{\sigma_1}, \dots, \sqrt{\sigma_{m-1}}, \underbrace{\sqrt{s_1/k}}_{q-m+1 \text{ times}})$$

### 4.2.3 FINDING ORTHONORMAL BASIS $\boldsymbol{X}$

We generated $\boldsymbol{X}$ by finding an orthonormal basis that was orthogonal to a vector $\boldsymbol{x}_0$ so that we could have $\boldsymbol{X}\boldsymbol{X}^\top = I - \boldsymbol{x}_0 \boldsymbol{x}_0^\top$. An efficient method of producing this basis is through Householder matrices $(\boldsymbol{x}_0, \boldsymbol{X}) = \boldsymbol{I} - 2\,\boldsymbol{v}\boldsymbol{v}^\top/||\boldsymbol{v}||^2$ where $\boldsymbol{v} = \boldsymbol{x}_0 - \boldsymbol{e}^{(1)}$ and $(\boldsymbol{x}_0, \boldsymbol{X})$ is a $k+1 \times k+1$ matrix with first column $\boldsymbol{x}_0$ and remaining columns $\boldsymbol{X}$ (Householder, 1958; user1551, 2013).

### 4.2.4 EFFICIENCY COMPARISONS TO STANDARD APPROACH

The OK/SKS methods require $\mathcal{O}((n_i + n_o + q)q^2)$ operations per sample and $\mathcal{O}(n_i n_o q)$ operations after collecting $B$ samples, giving an amortized cost of $\mathcal{O}((n_i + n_o + q)q^2 + n_i n_o q/B)$ operations per sample. Meanwhile, a standard approach expands the Kronecker sum at each sample, costing $\mathcal{O}(n_i n_o)$ operations per sample. If $q \ll B, n_i, n_o$ then the low rank method is superior to minibatch SGD in both memory and computational cost.

## 5 CONVEX CONVERGENCE

SKS introduces variance into the gradient estimates, so here we analyze the implications for online convex convergence. We analyze the case of strongly convex loss landscapes $f^t(\boldsymbol{w}^t)$ for flattened weight vector $\boldsymbol{w}^t$ and online sample $t$. In Appendix B, we show that with inverse squareroot learning rate, when the loss landscape Hessians satisfy $0 \prec c\boldsymbol{I} \preceq \nabla^2 f^t(\boldsymbol{w}^t)$ and under constraint (4) for the size of gradient errors $\boldsymbol{\varepsilon}^t$, where $\boldsymbol{w}^*$ is the optimal offline weight vector, the online regret (5) is sublinear in the number of online steps $T$. We can approximate $||\boldsymbol{\varepsilon}||$ and show that convex convergence is likely when (6) is satisfied in the biased, zero-variance case (equivalent to raw SVD, i.e., not applying Section 4.1.2), or when (7) is satisfied in the unbiased, minimum-variance case.

$$||\boldsymbol{\varepsilon}^t|| \le \frac{c}{2}||\boldsymbol{w}^t - \boldsymbol{w}^*|| \qquad (4) \qquad\qquad R(T) = \sum_{t=1}^T f^t(\boldsymbol{w}^t) - \sum_{t=1}^T f^t(\boldsymbol{w}^*) \qquad (5)$$

$$\sum_{i=1}^B \left(\sigma_q^{(t,i)}\right)^2 \le \frac{c^2}{4}||\boldsymbol{w}^t - \boldsymbol{w}^*||^2 \qquad (6) \qquad\qquad \sum_{i=1}^B \sigma_r^{(t,i)}\sigma_q^{(t,i)} \le \frac{c^2}{8}||\boldsymbol{w}^t - \boldsymbol{w}^*||^2 \qquad (7)$$

Equations (6, 7) suggest conditions under which fast convergence may be more or less likely and also point to methods for improving convergence. We discuss these in more detail in Appendix B.3.

## 5.1 CONVERGENCE EXPERIMENTS

We validate (4) with several linear regression experiments on a static input batch $\boldsymbol{X} \in \mathbb{R}^{1024 \times 100}$ and target $\boldsymbol{Y}_t \in \mathbb{R}^{256 \times 100}$. In Figure 1(a), Gaussian noise at different strengths (represented by different colors) is added to the true batch gradients at each update step. Notice that convergence slows significantly to the right of the dashed lines, which is the region where (4) no longer holds[4].

In Figure 1(b), we validate Equations (4, 6, 7) by testing the SVD and SKS cases with rank $r = 10$. In these particular experiments, SKS adds too much variance, causing it to operate to the right of the dashed lines. However, both SVD and SKS can be seen to reduce their variance as training progresses. In the case of SVD, it is able to continue training as it tracks the right dashed line.

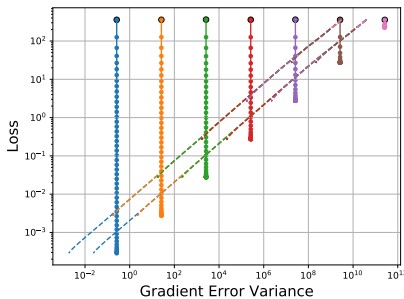 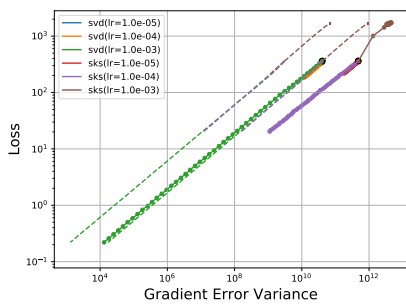

(a) True gradients with artificial noise      (b) SVD/SKS gradients over learning rates

Figure 1: In both plots, the solid line with markers plots the loss vs. gradient error variance (LHS of (4)) across 50 steps of SGD for several different setups. The left dashed line represents the RHS of (4) and the right dashed line is the RHS with $C$ instead of $c$.

## 6 IMPLEMENTATION DETAILS

**Quantization.** The NN is quantized in both the forward and backward directions with uniform power-of-2 quantization, where the clipping ranges are fixed at the start of training[5]. Weights are quantized to 8 bits between -1 and 1, biases to 16 bits between -8 and 8, activations to 8 bits between 0 and 2, and gradients to 8 bits between -1 and 1. Both the weights $\boldsymbol{W}$ and weight updates $\Delta \boldsymbol{W}$ are quantized to the same LSB so that weights cannot be used for accumulation beyond the fixed quantization dynamic range. This is in contrast to using high bitwidth (Zhou et al., 2016; Banner et al., 2018) or floating point accumulators. See Appendix D for more details on quantization.

**Gradient Max-Norming.** State-of-the-art methods in training, such as Adam (Kingma & Ba, 2014), use auxiliary memory per parameter to normalize the gradients. Unfortunately, we lack the memory budget to support these additional variables, especially if they must be updated every sample[6]. Instead, we propose dividing each gradient tensor by the maximum absolute value of its elements. This stabilizes the range of gradients across samples. See Appendix E for more details on gradient max-norming. In the experiments, we refer to this method as "max-norm" (opposite "no-norm").

**Streaming Batch Normalization.** Batch normalization (Ioffe & Szegedy, 2015) is a powerful technique for improving training performance which has been suggested to work by smoothing the loss landscape (Santurkar et al., 2018). We hypothesize that this may be especially helpful when parameters are quantized as in our case. However, in the online setting, we receive samples one-at-a-time rather than in batches. We therefore propose a streaming batch norm that uses moving average statistics rather than batch statistics as described in detail in Appendix F.

---

[4]As discussed in Appendix B.1, $B < n_i$, so we substitute $c$ in $c/2||\boldsymbol{w}^t - \boldsymbol{w}^*||$ with the minimum non-zero Eigenvalue of the Hessian $\tilde{c}$ when plotting the RHS of (4).

[5]Future work might look into how to change these clipping ranges, but this is beyond the scope of this paper.

[6]SKS could potentially approximate Adam. SKS on $\boldsymbol{a}^2, \boldsymbol{dz}^2, \boldsymbol{a}, \boldsymbol{dz}$ allows for a low-rank approximation of the variance of the gradients, however, this is unlikely to work well because of numerical stability (e.g., estimated variances might be negative).

# 7 EXPERIMENTS

## 7.1 ADAPTATION EXPERIMENTS

To test the effectiveness of SKS, experiments are performed on a representative CNN with four $3 \times 3$ convolution layers and two fully-connected layers. We generate "offline" and "online" datasets based on MNIST (see Appendix G), including one in which the statistical distribution shifts every 10k images. We then optimize an online SGD and rank-4 SKS model for fair comparison (see Appendix H). To see the importance of different training techniques, we run several ablations in Appendix I. Finally, we compare these different training schemes in different environments, meant to model real life. In these hypothetical scenarios, a model is first trained on the offline training set, and is then deployed to a number of devices at the edge that make supervised predictions (they make a prediction, then are told what the correct prediction would have been).

We present results on four hypothetical scenarios. First, a control case where both external/environment and internal/NVM drift statistics are exactly the same as during offline training. Second, a case where the input image statistical distribution shifts every 10k samples, selecting from augmentations such as spatial transforms and background gradients (see Section G). Third and fourth are cases where the NVM drifts from the programmed values, roughly modeling NVM memory degradation. In the third case, Gaussian noise is applied to the weights as if each weight was a single multi-level memory cell whose analog value drifted in a Brownian way. In the fourth case, random bit flips are applied as if each weight was represented by $b$ memory cells (see Appendix G for details). For each hypothetical scenario, we plot five different training schemes: pure quantized inference (no training), bias-only training, standard SGD training, SKS training, and SKS training with max-normed gradients. In SGD training and for training biases, parameters are updated at every step in an online fashion. These are seen as different colored curves in Figure 2.

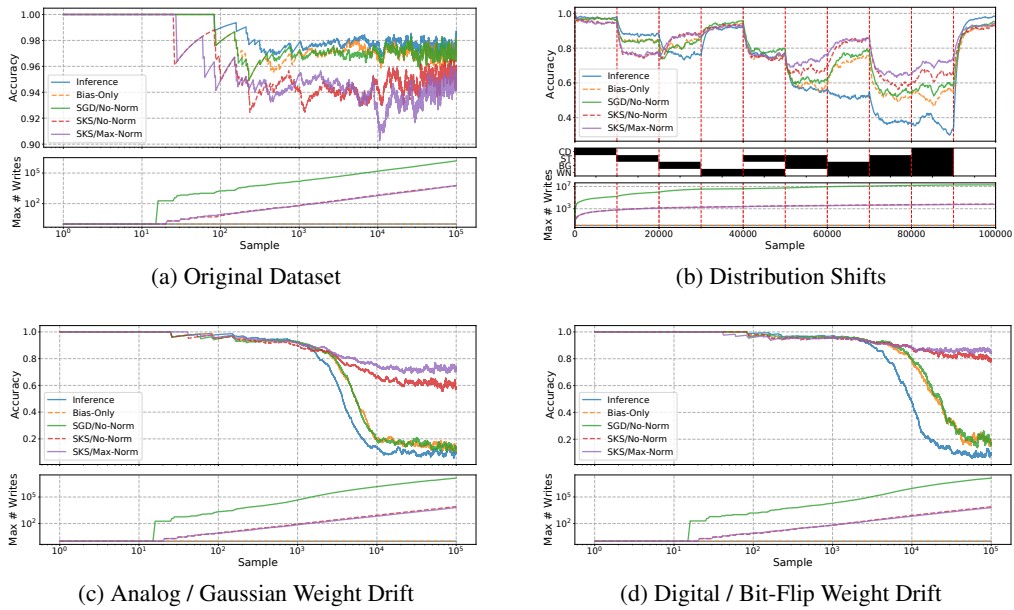

Figure 2: Adaptation of various training schemes over four different training environments (a) to (d). In each training environment, the top plot shows the exponential moving averages (0.999) of the per-sample online accuracy of the five training schemes, while the bottom plot shows the maximum number of updates applied to any given convolution or fully-connected kernel memory cell. For the distribution shifts in (b), the enabled augmentations at each contiguous 10k samples is shown (CD = class distribution, ST = spatial transforms, BG = background gradients, WN = white noise).

Inference does best in the control case, but does poorly in adaptation experiments. SGD doesn't improve significantly on bias-only training, likely because SGD cannot accumulate gradients less than a weight LSB. SKS, on the other hand, shows significant improvement, especially after several

thousand samples in the weight drift cases. Additionally, SKS shows about three orders of magnitude improvement compared to SGD in the worst case number of weight updates. Much of this reduction is due to the convolutions, where updates are applied at each pixel. However, reduction in fully-connected writes is still important because of potential energy savings. SKS/max-norm performs best in terms of accuracy across all environments and has similar weight update cost to SKS/no-norm.

## 7.2 TRANSFER LEARNING AND ALGORITHM COMPARISONS

To test the broader applicability of low rank training techniques, we run several experiments on ImageNet with ResNet-34 (Deng et al., 2009; He et al., 2016), a potentially realistic target for dense NVM inference on-chip. For ImageNet-size images, updating the low-rank approximation at each pixel quickly becomes infeasible, both because of the single-threaded nature of the algorithm, and because of the increased variance of the estimate at larger batch sizes. Instead, we focus on training the final layer weights ($1000 \times 512$). ResNet-34 weights are initialized to those from Paszke et al. (2017) and the convolution layers are used to generate feature vectors for 10k ImageNet training images[7], which are quantized and fed to a one-layer quantized[8] neural network. To speed up experiments, the layer weights are initialized to the pretrain weights, modulated by random noise that causes inference top-1 accuracy to fall to $52.7\% \pm 0.9\%$. In Table 1, we see that the unbiased SKS has the strongest recovery accuracies, although biased SVD also does quite well. The high-variance UORO and true SGD have weak or non-existent recoveries.

Table 1: Accuracy recovery beyond inference (%, mean with standard deviation from 5 random seeds) between different algorithms (all with max-norm; effective batch size $B = 100$ if applicable), tested at different ranks ($r$), and learning rates ($\eta$). Optimal learning rates are bolded.

| Algorithm | $\eta$
$r$ | 0.003 | 0.010 | 0.030 | 0.100 | 0.300 |
|---|---|---|---|---|---|---|
| SGD | - | $+0.3 \pm 0.2$ | $+0.3 \pm 0.2$ | $+0.3 \pm 0.2$ | $\mathbf{+0.9 \pm 0.2}$ | $-3.9 \pm 0.8$ |
| UORO | 1 | $\mathbf{+0.4 \pm 0.2}$ | $+0.3 \pm 0.4$ | $-1.8 \pm 0.9$ | $-7.6 \pm 1.6$ | $-31.7 \pm 1.6$ |
| SVD | 1 | $+1.9 \pm 0.2$ | $\mathbf{+5.8 \pm 1.0}$ | $-3.4 \pm 1.0$ | $-19.4 \pm 0.9$ | $-40.7 \pm 1.1$ |
| | 2 | $+1.4 \pm 0.4$ | $\mathbf{+6.5 \pm 0.7}$ | $+6.3 \pm 0.6$ | $-5.2 \pm 0.9$ | $-36.3 \pm 0.9$ |
| | 4 | $+1.3 \pm 0.4$ | $\mathbf{+6.5 \pm 0.7}$ | $+5.2 \pm 0.8$ | $-3.3 \pm 1.0$ | $-33.8 \pm 0.8$ |
| | 8 | $+1.4 \pm 0.3$ | $\mathbf{+5.6 \pm 0.8}$ | $+4.3 \pm 0.9$ | $-2.4 \pm 1.0$ | $-32.8 \pm 0.9$ |
| SKS | 1 | $\mathbf{+0.3 \pm 0.2}$ | $+0.3 \pm 0.2$ | $-0.7 \pm 0.4$ | $-2.7 \pm 1.7$ | $-26.5 \pm 2.6$ |
| | 2 | $+0.3 \pm 0.2$ | $+0.4 \pm 0.3$ | $-0.1 \pm 0.4$ | $\mathbf{+1.3 \pm 0.9}$ | $-12.9 \pm 1.1$ |
| | 4 | $+0.4 \pm 0.2$ | $+0.6 \pm 0.2$ | $+1.9 \pm 0.3$ | $\mathbf{+8.0 \pm 1.1}$ | $-5.1 \pm 1.1$ |
| | 8 | $+0.4 \pm 0.2$ | $+1.1 \pm 0.2$ | $+3.3 \pm 0.7$ | $\mathbf{+4.8 \pm 1.5}$ | $-15.8 \pm 1.7$ |

## 8 CONCLUSION

We demonstrated the potential for SKS to solve the major challenges facing online training on NVM-based edge devices: low write density and low auxiliary memory. SKS is a computationally-efficient, memory-light algorithm capable of decoupling batch size from auxiliary memory, allowing larger effective batch sizes, and consequently lower write densities. Additionally, we noted that SKS may allow for training under severe weight quantization constraints as rudimentary gradient accumulations are handled by the $\boldsymbol{L}, \boldsymbol{R}$ matrices, which can have high bitwidths (as opposed to SGD, which may squash small gradients to 0). We found expressions for when SKS might have better convergence properties. Across a variety of online adaptation problems and a large-scale transfer learning demonstration, SKS was shown to match or exceed the performance of SGD while using a small fraction of the number of updates. Finally, we suspect that these techniques could be applied to a broader range of problems. Auxiliary memory minimization may be analogous to communication minimization in training strategies such as federated learning, where gradient compression is important.

---

[7]The decision to use training data is deliberate, however experiments on out-of-sample images, such as Recht et al. (2019) show similar behavior.

[8]Quantization ranges are chosen to optimize accuracy and are different from those in Section 7.1.

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

## A  SKS ALGORITHM

---

**Algorithm 1** Streaming Kronecker Sum Approximation

---

**State:** $\boldsymbol{Q}_L \in \mathbb{R}^{n_o \times q}$; $\boldsymbol{Q}_R \in \mathbb{R}^{n_i \times q}$; $\boldsymbol{c}_x \in \mathbb{R}^{q \times 1}$
**Input:** $\boldsymbol{dz}^{(i)} \in \mathbb{R}^{n_o \times 1}$; $\boldsymbol{a}^{(i)} \in \mathbb{R}^{n_i \times 1}$ for $i \in [1, B]$
**for** $i = 1 \ldots B$ **do**
  {Modified Gram-Schmidt.}
  $\boldsymbol{c}_L, \boldsymbol{c}_R \leftarrow 0^{q \times 1}$
  **for** $j = 1 \ldots r$ **do**
    $c_{L,j} \leftarrow \boldsymbol{Q}_{L,j} \cdot \boldsymbol{dz}^{(i)}$;   $\boldsymbol{dz}^{(i)} \leftarrow \boldsymbol{dz}^{(i)} - c_{L,j} \cdot \boldsymbol{Q}_{L,j}$
    $c_{R,j} \leftarrow \boldsymbol{Q}_{R,j} \cdot \boldsymbol{a}^{(i)}$;   $\boldsymbol{a}^{(i)} \leftarrow \boldsymbol{a}^{(i)} - c_{L,j} \cdot \boldsymbol{Q}_{L,j}$
  **end for**
  $c_{L,q} \leftarrow \|\boldsymbol{dz}^{(i)}\|$;   $\boldsymbol{Q}_{L,q} \leftarrow \boldsymbol{dz}^{(i)}/c_{L,q}$
  $c_{R,q} \leftarrow \|\boldsymbol{a}^{(i)}\|$;   $\boldsymbol{Q}_{R,q} \leftarrow \boldsymbol{a}^{(i)}/c_{R,q}$
  {Generate $\boldsymbol{C}$ and find its SVD.}
  $\boldsymbol{C} \leftarrow \boldsymbol{c}_L \boldsymbol{c}_R^\top + \mathrm{diag}(\boldsymbol{c}_x)$
  $\boldsymbol{U}_C \cdot \mathrm{diag}(\boldsymbol{\sigma}) \cdot \boldsymbol{V}_C^\top \leftarrow \mathrm{SVD}(\boldsymbol{C})$
  {Minimum-variance unbiased estimator for $\boldsymbol{\Sigma}$.}
  $m \leftarrow \min\ j$ s.t. $(q - j)\sigma_j \leq \sum_{\ell=j}^q \sigma_\ell$
  $s_1 \leftarrow \sum_{i=m}^q \sigma_i$,   $k \leftarrow q - m$
  $\boldsymbol{v} \leftarrow \sqrt{1 - k/s_1} \cdot \boldsymbol{\sigma}_{[m:]} - \boldsymbol{e}^{(1)}$
  $\mathbf{s} \leftarrow \{-1, 1\}^{(k+1) \times 1}$ {Ind. uniform random signs.}
  $\boldsymbol{X}_s \leftarrow \left(\boldsymbol{I} + (\mathbf{s} \odot \boldsymbol{v})(\boldsymbol{v}/v_1)^\top\right)_{[2:]}$ {Householder.}
  {QR-factorization of $\tilde{\boldsymbol{\Sigma}}_L$.}
  $\boldsymbol{Q}_x \leftarrow \begin{bmatrix} \boldsymbol{I} & 0 \\ 0 & \boldsymbol{X}_s \end{bmatrix} \in \mathbb{R}^{q \times r}$
  $\boldsymbol{c}_x \leftarrow (\sigma_1, \ldots, \sigma_{m-1}, \underbrace{s_1/k, \ldots, s_1/k}_{q-m+1\ \text{times}})$
  {Update the first $r$ columns of $\boldsymbol{Q}_L, \boldsymbol{Q}_R$.}
  $\boldsymbol{Q}_{L[:r]} \leftarrow \boldsymbol{Q}_L \cdot \boldsymbol{U}_C \cdot \boldsymbol{Q}_x$
  $\boldsymbol{Q}_{R[:r]} \leftarrow \boldsymbol{Q}_R \cdot \boldsymbol{V}_C \cdot \boldsymbol{Q}_x$
**end for**
{Compute final $\tilde{\boldsymbol{L}}, \tilde{\boldsymbol{R}}$ where $\nabla_{\boldsymbol{W}} \mathcal{L} \approx \tilde{\boldsymbol{L}}\tilde{\boldsymbol{R}}^\top$.}
$\tilde{\boldsymbol{L}} \leftarrow \left(\boldsymbol{Q}_L \cdot \mathrm{diag}(\sqrt{\boldsymbol{c}_x})\right)_{[:r]}$
$\tilde{\boldsymbol{R}} \leftarrow \left(\boldsymbol{Q}_R \cdot \mathrm{diag}(\sqrt{\boldsymbol{c}_x})\right)_{[:r]}$

---

## B  CONVEX CONVERGENCE

In this section we will attempt to bound the regret (defined below) of an SGD algorithm using noisy SKS estimates $\tilde{\boldsymbol{g}} = \boldsymbol{g} + \boldsymbol{\varepsilon}$ in the convex setting, where $\boldsymbol{g}$ are the true gradients and $\boldsymbol{\varepsilon}$ are the errors introduced by the low rank SKS approximation. Here, $\boldsymbol{g}$ is a vector of size $N$ and can be thought of as a flattened/concatenated version of the gradient tensors (e.g., $N = n_i \cdot n_o$).

Our proof follows the proof in Zinkevich (2003). We define $\mathbb{F}$ as the convex feasible set (valid settings for our weight tensors) and assume that $\mathbb{F}$ is bounded with $D = \max_{\boldsymbol{w}, \boldsymbol{v} \in \mathbb{F}} \|\boldsymbol{w} - \boldsymbol{v}\|$ being the maximum distance between two elements of $\mathbb{F}$. Further, assume a batch $t$ of $B$ samples out of $T$ total batches corresponds to a loss landscape $f^t(\boldsymbol{w}^t)$ that is strongly convex in weight parameters $\boldsymbol{w}^t$, so there are positive constants $C \geq c > 0$ such that $c\boldsymbol{I} \preceq \nabla^2 f^t(\boldsymbol{w}^t) \preceq C\boldsymbol{I}$ for all $t$ (Boyd & Vandenberghe, 2004, Section 9.3). We define regret as $R(T) = \sum_{t=1}^T f^t(\boldsymbol{w}^t) - \sum_{t=1}^T f^t(\boldsymbol{w}^*)$ where $\boldsymbol{w}^* = \mathrm{argmin}_{\boldsymbol{w}} \sum_{t=1}^T f^t(\boldsymbol{w})$ (i.e., it is an optimal offline minimizer of $f^1, \cdots, f^T$).

The gradients seen during SGD are $\boldsymbol{g}^t = \nabla f^t(\boldsymbol{w}^t)$ and we assume they are bounded by $G = \max_{\boldsymbol{w} \in \mathbb{F}, t \in [1,T]} ||\nabla f^t(\boldsymbol{w})||$. We also assume errors are bounded by $\mathcal{E} = \max_{t \in [1,T]} ||\boldsymbol{\varepsilon}^t||$. Therefore, $\max_{t \in [1,T]} ||\tilde{\boldsymbol{g}}^t|| \leq \max_{t \in [1,T]} ||\boldsymbol{g}^t|| + ||\boldsymbol{\varepsilon}^t|| \leq G + \mathcal{E}$ by the triangle inequality.

**Theorem 1.** *Assume SKS-based SGD is applied with learning rate $\eta_t = 1/\sqrt{t}$. Then, under the additional constraint $\boldsymbol{g}^t \cdot (\boldsymbol{w}^t - \boldsymbol{w}^*) - \frac{c}{2} ||\boldsymbol{w}^t - \boldsymbol{w}^*||_2^2 \leq \tilde{\boldsymbol{g}}^t \cdot (\boldsymbol{w}^t - \boldsymbol{w}^*)$, we have sublinear regret:*

$$R(T) \leq \frac{D^2}{2} \sqrt{T} + (G + \mathcal{E})^2 \left( \sqrt{T} - 1/2 \right)$$

*Proof.* From strong convexity $c\boldsymbol{I} \preceq \nabla^2 f^t(\boldsymbol{w}^t) \preceq C\boldsymbol{I}$ for all $t$,

$$f^t(\boldsymbol{w}) + \boldsymbol{g}^t \cdot (\boldsymbol{v} - \boldsymbol{w}) + \frac{c}{2} ||\boldsymbol{v} - \boldsymbol{w}||_2^2 \leq f^t(\boldsymbol{v}) \quad \text{for all } \boldsymbol{v} \tag{8}$$

In particular, if we consider $\boldsymbol{v} = \boldsymbol{w}^*$ and rearrange,

$$f^t(\boldsymbol{w}) - f^t(\boldsymbol{w}^*) \leq \boldsymbol{g}^t \cdot (\boldsymbol{w} - \boldsymbol{w}^*) - \frac{c}{2} ||\boldsymbol{w} - \boldsymbol{w}^*||_2^2$$
$$f^t(\boldsymbol{w}) - f^t(\boldsymbol{w}^*) \leq \tilde{\boldsymbol{g}}^t \cdot (\boldsymbol{w}^t - \boldsymbol{w}^*) \tag{9}$$

Consider a gradient update $\boldsymbol{w}^{t+1} = P_{\mathbb{F}}(\boldsymbol{w}^t - \eta_t \tilde{\boldsymbol{g}}^t)$, where $P_{\mathbb{F}}$ projects the update back to $\mathbb{F}$. Then,

$$\begin{aligned}
||\boldsymbol{w}^{t+1} - \boldsymbol{w}^*||_2^2 &= ||P(\boldsymbol{w}^t - \eta_t \tilde{\boldsymbol{g}}^t) - \boldsymbol{w}^*||_2^2 \leq ||\boldsymbol{w}^t - \eta_t \tilde{\boldsymbol{g}}^t - \boldsymbol{w}^*||_2^2 \\
&= ||\boldsymbol{w}^t - \boldsymbol{w}^*||_2^2 - 2\eta_t (\boldsymbol{w}^t - \boldsymbol{w}^*) \cdot \tilde{\boldsymbol{g}}^t + \eta_t^2 ||\tilde{\boldsymbol{g}}^t||_2^2 \\
&\leq ||\boldsymbol{w}^t - \boldsymbol{w}^*||_2^2 - 2\eta_t (\boldsymbol{w}^t - \boldsymbol{w}^*) \cdot \tilde{\boldsymbol{g}}^t + \eta_t^2 (G + \mathcal{E})^2 \\
\tilde{\boldsymbol{g}}^t \cdot (\boldsymbol{w}^t - \boldsymbol{w}^*) &\leq \frac{1}{2\eta_t} \left( ||\boldsymbol{w}^t - \boldsymbol{w}^*||_2^2 - ||\boldsymbol{w}^{t+1} - \boldsymbol{w}^*||_2^2 \right) + \frac{\eta_t}{2} (G + \mathcal{E})^2
\end{aligned} \tag{10}$$

From (9, 10),

$$f^t(\boldsymbol{w}^t) - f^t(\boldsymbol{w}^*) \leq \frac{1}{2\eta_t} \left( ||\boldsymbol{w}^t - \boldsymbol{w}^*||_2^2 - ||\boldsymbol{w}^{t+1} - \boldsymbol{w}^*||_2^2 \right) + \frac{\eta_t}{2} (G + \mathcal{E})^2 \tag{11}$$

We now bound the regret:

$$\begin{aligned}
R(T) &= \sum_{t=1}^T \left[ f^t(\boldsymbol{w}^t) - f^t(\boldsymbol{w}^*) \right] \\
&\leq \sum_{t=1}^T \left[ \frac{1}{2\eta_t} \left( ||\boldsymbol{w}^t - \boldsymbol{w}^*||_2^2 - ||\boldsymbol{w}^{t+1} - \boldsymbol{w}^*||_2^2 \right) + \frac{\eta_t}{2} (G + \mathcal{E})^2 \right] \\
&= \frac{||\boldsymbol{w}^1 - \boldsymbol{w}^*||_2^2}{2\eta_1} - \frac{||\boldsymbol{w}^{T+1} - \boldsymbol{w}^*||_2^2}{2\eta_T} + \frac{1}{2} \sum_{t=2}^T \left( \frac{1}{\eta_t} - \frac{1}{\eta_{t-1}} \right) ||\boldsymbol{w}^t - \boldsymbol{w}^*||_2^2 + \frac{(G + \mathcal{E})^2}{2} \sum_{t=1}^T \eta_t \\
&\leq \frac{||\boldsymbol{w}^1 - \boldsymbol{w}^*||_2^2}{2\eta_1} + \frac{1}{2} \sum_{t=2}^T \left( \frac{1}{\eta_t} - \frac{1}{\eta_{t-1}} \right) ||\boldsymbol{w}^t - \boldsymbol{w}^*||_2^2 + \frac{(G + \mathcal{E})^2}{2} \sum_{t=1}^T \eta_t \\
&\leq \frac{D^2}{2\eta_1} + \frac{1}{2} \sum_{t=2}^T \left( \frac{1}{\eta_t} - \frac{1}{\eta_{t-1}} \right) D^2 + \frac{(G + \mathcal{E})^2}{2} \sum_{t=1}^T \eta_t \\
&= \frac{D^2}{2\eta_T} + \frac{(G + \mathcal{E})^2}{2} \sum_{t=1}^T \eta_t
\end{aligned} \tag{12}$$

If $\eta_t = 1/\sqrt{t}$, then $\sum_{t=1}^{T} \eta_t \leq 2\sqrt{T} - 1$ (Zinkevich, 2003), so from (12),

$$R(T) \leq \frac{D^2}{2}\sqrt{T} + (G + \mathcal{E})^2 \left(\sqrt{T} - 1/2\right) \tag{13}$$

$\square$

This is a sublinear regret and therefore, average regret $R(T)/T$ is bounded above by 0 in the limit as $T \to \infty$. To achieve this result, we constrained $\boldsymbol{g}^t \cdot (\boldsymbol{w}^t - \boldsymbol{w}^*) - \frac{c}{2}||\boldsymbol{w}^t - \boldsymbol{w}^*||_2^2 \leq \tilde{\boldsymbol{g}}^t \cdot (\boldsymbol{w}^t - \boldsymbol{w}^*)$. We now examine sufficient conditions for this inequality to be satisfied.

$$\boldsymbol{g}^t \cdot (\boldsymbol{w}^t - \boldsymbol{w}^*) - \frac{c}{2}||\boldsymbol{w}^t - \boldsymbol{w}^*||_2^2 \leq \tilde{\boldsymbol{g}}^t \cdot (\boldsymbol{w}^t - \boldsymbol{w}^*)$$
$$\varepsilon^t \cdot (\boldsymbol{w}^* - \boldsymbol{w}^t) \leq \frac{c}{2}||\boldsymbol{w}^t - \boldsymbol{w}^*||_2^2 \tag{14}$$

Since $\varepsilon^t \cdot (\boldsymbol{w}^* - \boldsymbol{w}^t) \leq ||\varepsilon^t|| \cdot ||\boldsymbol{w}^* - \boldsymbol{w}^t||$ by Cauchy-Schwarz, it is sufficient for:

$$||\varepsilon^t|| \cdot ||\boldsymbol{w}^* - \boldsymbol{w}^t|| \leq \frac{c}{2}||\boldsymbol{w}^t - \boldsymbol{w}^*||_2^2$$
$$||\varepsilon^t|| \leq \frac{c}{2}||\boldsymbol{w}^t - \boldsymbol{w}^*|| \tag{15}$$

## B.1 CONSIDERATIONS FOR RANK DEFICIENT HESSIANS

In the preceding proof, we assumed $c > 0$. However, it is common for this to not hold. For example, in linear regression, where $c = \lambda_{min}(\boldsymbol{X}\boldsymbol{X}^\top)$ for sample input $\boldsymbol{X} \in \mathbb{R}^{(n_i \times B)}$ (Haykin, 2014, Chapter 4.3), if $B < n_i$ then $c = 0$. We can modify (15) to handle this case. Let $\tilde{c}$ be the minimum non-zero Eigenvalue of $\boldsymbol{X}\boldsymbol{X}^\top$ and let $\tilde{\boldsymbol{w}} \in \mathbb{R}^B$ represent $\boldsymbol{w} \in \mathbb{R}^N$ in the Eigenbasis of $\boldsymbol{X}\boldsymbol{X}^\top$. Then (15) becomes:

$$||\varepsilon^t|| \leq \frac{\tilde{c}}{2}||\tilde{\boldsymbol{w}}^t - \tilde{\boldsymbol{w}}^*|| \tag{16}$$

## B.2 ESTIMATES FOR THE SKS ERROR

We can estimate the SKS error $||\varepsilon^t||$ in both the biased and unbiased cases. For the **biased, zero-variance case**, we get rid of the lowest singular value (out of $q$ singular values) as we see each sample. Thus, at a given sample $i$, the error is $\sigma_q^{(i)} \cdot \boldsymbol{Q}_{L,q}^{(i)} \cdot \boldsymbol{Q}_{R,q}^{(i)}$ and the average squared error is $(1/N)\sigma_q^{(i)2}$. We can treat this as a per-element variance. If these smallest singular components are uncorrelated from sample to sample, then the variances add:

$$\sigma_\varepsilon^2 \approx \frac{1}{N}\sum_{i=1}^{B}\left(\sigma_q^{(i)}\right)^2 \tag{17}$$

For the **unbiased, minimum-variance case**, Theorem A.4 from Benzing et al. (2019) states that the minimum variance is $s_1^2/k + s_2$ where $s_1 = \sum_{i=m}^{q}\sigma_i$, $s_2 = \sum_{i=m}^{q}\sigma_i^2$, and $k, m$ are as defined in Section 4.1.2. Since $m$ is chosen to minimize variance, we can upper bound the variance by choosing $m = r$ and therefore $k = 1$, $s_1 = \sigma_r + \sigma_q$, and $s_2 = \sigma_r^2 + \sigma_q^2$. Empirically, this tends to be a good approximation. Then, the average per-element variance added at sample $i$ is approximately $(2/N)\left(\sigma_r^{(i)}\sigma_q^{(i)}\right)$. Assuming errors between samples are uncorrelated, this leads to a total variance:

$$\sigma_\varepsilon^2 \approx \frac{2}{N}\sum_{i=1}^{B}\sigma_r^{(i)}\sigma_q^{(i)} \tag{18}$$

For either case, $||\boldsymbol{\varepsilon}||^2 \approx N\sigma_\varepsilon^2$. For the $t$-th batch and $i$-th sample, we denote $\sigma_q^{(t,i)}$ as the $q$-th singular value. For simplicity, we focus on the biased, zero-variance case (the unbiased case is similar). From (15), an approximately sufficient condition for sublinear-regret convergence is:

$$\sum_{i=1}^{B} \left(\sigma_q^{(t,i)}\right)^2 \le \frac{c^2}{4}||\boldsymbol{w}^t - \boldsymbol{w}^*||^2 \tag{19}$$

### B.3 Discussion on Convergence

Equation (19) suggests that as $\boldsymbol{w}^t \to \boldsymbol{w}^*$, the constraints for achieving sublinear-regret convergence become more difficult to maintain. However, in practice this may be highly problem-dependent as the $\sigma_q$ will also tend to decrease near optimal solutions. To get a better sense of the behavior of the left-hand side of (19), suppose that:

$$\sum_{i=1}^{B} \left(\sigma_q^{(t,i)}\right)^2 \approx \sum_{i=q}^{B} \left(\sigma_i(\boldsymbol{G}^t)\right)^2 \le \sum_{i=1}^{B} \left(\sigma_i(\boldsymbol{G}^t)\right)^2 = ||\boldsymbol{G}^t||_F^2$$

where $\boldsymbol{G}^t = \nabla_{\boldsymbol{W}^t} f^t(\boldsymbol{W}^t) \in \mathbb{R}^{(n_o \times n_i)}$ are the matrix weight $\boldsymbol{W}^t$ gradients at batch $t$ and $||\cdot||_F$ is a Frobenius norm. We therefore expect both the left (proportional to $||\boldsymbol{G}^t||_F^2$) and the right (proportional to $||\boldsymbol{w}^t - \boldsymbol{w}^*||^2$) of (19) to decrease during training as $\boldsymbol{w}^t \to \boldsymbol{w}^*$. This behavior is in fact what is seen in Figure 1(b). If achieving convergence is found to be difficult, (19) provides some insight for convergence improvement methods.

One solution is to reduce batch size $B$ to satisfy the inequality as necessary. This minimizes the weight updates during more repetitive parts of training while allowing dense weight updates (possibly approaching standard SGD with small batch sizes) during more challenging parts of training.

Another solution is to reduce $\sigma_q$. One way to do this is to increase the rank $r$ so that the spectral energy of the updates are spread across more singular components. There may be alternate approaches based on conditioning the inputs to shape the distribution of singular values in a beneficial way.

A third method is to focus on $c$, the lower bound on curvature of the convex loss functions. Perhaps a technique such as weight regularization can increase $c$ by adding constant curvature in all Eigen-directions of the loss function Hessian (although this may also increase the LHS of (19)). Alternatively, perhaps low-curvature Eigen-directions are less important for loss minimization, allowing us to raise the $c$ that we effectively care about. This latter approach requires no particular action on our part, except the recognition that fast convergence may only be guaranteed for high-curvature directions. This is exemplified in Figure 1(b), where we can see SVD track the curve for $C$ more so than $c$.

Finally, we note that this analysis focuses solely on the errors introduced by a floating-point version of SKS. Quantization noise can add additional error into the $\boldsymbol{\varepsilon}^t$ term. We expect this to add a constant offset to the LHS of (19). For a weight LSB $\Delta$, quantization noise has variance $\Delta^2/12$, so we desire:

$$N\frac{\Delta^2}{12} + \sum_{i=1}^{B} \left(\sigma_q^{(t,i)}\right)^2 \le \frac{c^2}{4}||\boldsymbol{w}^t - \boldsymbol{w}^*||^2 \tag{20}$$

## C    Kronecker Sums in Neural Network Layers

### C.1    Dense Layer

A dense or fully-connected layer transforms an input $\boldsymbol{a} \in \mathbb{R}^{n_i \times 1}$ to an intermediate $\boldsymbol{z} = \boldsymbol{W} \cdot \boldsymbol{a} + \boldsymbol{b}$ to an output $\boldsymbol{y} = \sigma(\boldsymbol{z}) \in \mathbb{R}^{n_o \times 1}$ where $\sigma$ is a non-linear activation function. Gradients of the loss function with respect to the weight parameters can be found as:

$$\nabla_{\boldsymbol{W}} \mathcal{L} = \underbrace{(\nabla_{\boldsymbol{z}} \mathcal{L})}_{\boldsymbol{dz}} \odot \underbrace{(\nabla_{\boldsymbol{W}} \boldsymbol{z})}_{\boldsymbol{a}^\top} = \boldsymbol{dz} \otimes \boldsymbol{a} \tag{21}$$

which is exactly the per-sample Kronecker sum update we saw in linear regression. Thus, at every training sample, we can add $(dz^{(i)} \otimes a^{(i)})$ to our low rank estimate with SKS.

## C.2 Convolutional Layer

A convolutional layer transforms an input feature map $\mathbf{A} \in \mathbb{R}^{h_{in} \times w_{in} \times c_{in}}$ to an intermediate feature map $\mathbf{Z} = \mathbf{W}_{kern} * \mathbf{A} + \boldsymbol{b} \in \mathbb{R}^{h_{out} \times w_{out} \times c_{out}}$ through a 2D convolution $*$ with weight kernel $\mathbf{W}_{kern} \in \mathbb{R}^{c_{out} \times k_h \times k_w \times c_{in}}$. Then it computes an output feature map $y = \sigma(z)$ where $\sigma$ is a non-linear activation function.

Convolutions can be interpreted as matrix multiplications through the im2col operation which converts the input feature map $\mathbf{A}$ into a matrix $\boldsymbol{A}_{col} \in \mathbb{R}^{(h_{out}w_{out}) \times (k_h k_w c_{in})}$ where the $i^{\text{th}}$ row is a flattened version of the sub-tensor of $a$ which is dotted with $\mathbf{W}_{kern}$ to produce the $i^{\text{th}}$ pixel of the output feature map (Ren & Xu, 2015). We can multiply $\boldsymbol{A}_{col}$ by a flattened version of the kernel, $\boldsymbol{W} \in \mathbb{R}^{c_{out} \times (k_h h_w c_{in})}$ to perform the $\mathbf{W}_{kern} * \mathbf{A}$ convolution operation with a matrix multiplication. Under the matrix multiplication interpretation, weight gradients can be represented as:

$$\nabla_{\boldsymbol{W}} \mathcal{L} = \underbrace{(\nabla_{\boldsymbol{Z}_{col}} \mathcal{L})}_{d\boldsymbol{Z}_{col}^{\top}} \odot \underbrace{(\nabla_{\boldsymbol{W}} \boldsymbol{Z})}_{\boldsymbol{A}_{col}} = \sum_{i=1}^{h_{out}w_{out}} d\boldsymbol{Z}_{col,i}^{\top} \otimes \boldsymbol{A}_{col,i}^{\top} \tag{22}$$

which is the same as $h_{out}w_{out}$ Kronecker sum updates. Thus, at every output pixel $j$ of every training sample $i$, we can add $(d\boldsymbol{Z}_{col,j}^{(i)\top} \otimes \boldsymbol{A}_{col,j}^{(i)\top})$ to our low rank estimate with SKS.

Note that while we already save an impressive factor of $B/q$ in memory when computing gradients for the dense layer, we save a much larger factor of $Bh_{out}w_{out}/q$ in memory when computing gradients for the convolution layers, making the low rank training technique even more crucial here.

However, some care must be taken when considering activation memory for convolutions. For compute-constrained edge devices, image dimensions may be small and result in minimal intermediate feature map memory requirements. However, if image dimensions grow substantially, activation memory could dominate compared to weight storage. Clever dataflow strategies may provide a way to reduce intermediate activation storage even when performing backpropagation[9].

## D Hardware Quantization Model

In a real device, operations are expected to be performed in fixed point arithmetic. Therefore, all of our training experiments are conducted with quantization in the loop. Our model for quantization is shown in Figure 3. The green arrows describe the forward computation. Ignoring quantization for a moment, we would have $a^{\ell} = \text{ReLU}\left(\alpha^{\ell} W^{\ell} * a^{\ell-1} + b^{\ell}\right)$, where $*$ can represent either a convolution or a matrix multiply depending on the layer type and $\alpha^{\ell}$ is the closest power-of-2 to He initialization (He et al., 2015). For quantization, we rely on four basic quantizers: $Qw, Qb, Qa, Qg$, which describe weight quantization, bias and intermediate accumulator quantization, activation quantization, and gradient quantization, respectively. All quantizers use fixed clipping ranges as depicted and quantize uniformly within those ranges to the specified bitwidths.

In the backward pass, follow the orange arrows from $\boldsymbol{\delta}^{\ell}$. Backpropagation follows standard backpropagation rules including using the straight-through estimator (Bengio et al., 2013) for quantizer gradients. However, because we want to perform training on edge devices, these gradients must themselves be quantized. The first place this happens is after passing backward through the ReLU derivitive. The other two places are before feeding back into the network parameters $\boldsymbol{W}^{\ell}, \boldsymbol{b}^{\ell}$, so that $\boldsymbol{W}^{\ell}, \boldsymbol{b}^{\ell}$ cannot be used to accumulate values smaller than their LSB. Finally, instead of deriving $\Delta \boldsymbol{W}^{\ell}$ from a backward pass through the $*$ operator, the SKS method is used.

---

[9]For example, one could compute just a sliding window of rows of every feature map, discarding earlier rows as later rows are computed, resulting in a square-root reduction of activation memory. To incorporate backpropagation, compute the forward pass once fully, then compute the forward pass again, as well as the backward pass using the sliding window approach in both directions.

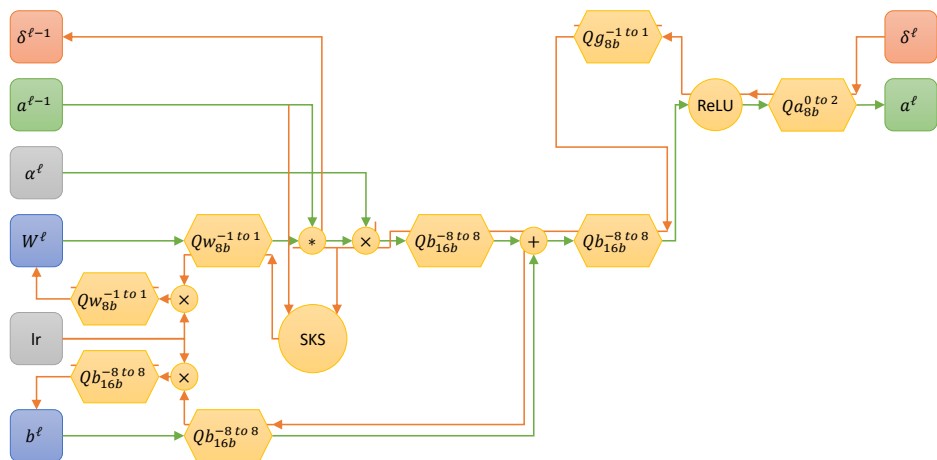

Figure 3: Signal flow graph for a forward and backward quantized convolutional or dense layer.

SKS collects $a^{\ell-1}, dz^\ell$ for many samples before computing the approximate $\Delta \tilde{W}^\ell$. It accumulates information in two low rank matrices $L, R$ which are themselves quantized to 16 bits with clipping ranges determined dynamically by the max absolute value of elements in each matrix. While SKS accumulates for $B$ samples, leading to a factor of $B$ reduction in the rate of updates to $W^\ell$, $b^\ell$ is updated at every sample. This is feasible in hardware because $b^\ell$ is small enough to be stored in more expensive forms of memory that have superior endurance and write power performance.

Because of the coarse weight LSB size, weight gradients may be consistently quantized to 0, preventing them from accumulating. To combat this, we only apply an update if a minimum update density $\rho_{\min} = 0.01$ would be achieved, otherwise we continue accumulating samples in $L$ and $R$, which have much higher bitwidths. When an update does finally happen, the "effective batch size" will be a multiple of $B$ and we increase the learning rate correspondingly. In the literature, a linear scaling rule is suggested (see Goyal et al. (2017)), however we empirically find square-root scaling works better (see Appendix H).

# E    GRADIENT MAX-NORMING

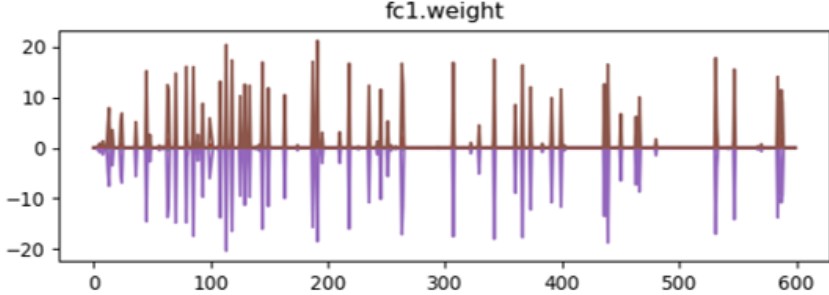

Figure 4: Maximum magnitude of weight gradients versus training step for standard SGD on a CNN trained on MNIST.

Figure 4 plots the magnitude of gradients seen in a weight tensor over training steps. One apparent property of these gradients is that they have a large dynamic range, making them difficult to quantize. Even when looking at just the spikes, they assume a wide range of magnitudes. One potential method of dealing with this dynamic range is to scale tensors so that their max absolute element is 1 (similar to a per-tensor AdaMax (Kingma & Ba, 2014) or Range Batch-Norm (Banner et al., 2018) applied to gradients). Optimizers such as Adam, which normalize by gradient variance, provide a justification for why this sort of scaling might work well, although they work at a per-element

rather than per-tensor level. We choose max-norming rather than variance-based norming because the former is easier computational and potentially more ammenable to quantization. However, a problem with the approach of normalizing tensors independently at each sample is that noise might be magnified during regions of quiet as seen in the Figure. What we therefore propose is normalization by the maximum of both the current max element and a moving average of the max element.

Explicitly, max-norm takes two parameters - a decay factor $\beta = 0.999$ and a gradient floor $\varepsilon = 10^{-4}$ and keeps two state variables - the number of evaluations $k := 0$ and the current maximum moving average $x_{mv} := \varepsilon$. Then for a given input $x$, max-norm modifies its internal state and returns $x_{norm}$:

$$k := k + 1$$
$$x_{max} := \max(|\boldsymbol{x}|) + \varepsilon$$
$$x_{mv} := \beta \cdot x_{mv} + (1 - \beta) \cdot x_{max}$$
$$\tilde{x}_{mv} := \frac{x_{mv}}{1 - \beta^k}$$
$$\boldsymbol{x}_{norm} := \frac{\boldsymbol{x}}{\max(x_{max}, \tilde{x}_{mv})}$$

## F  STREAMING BATCH NORMALIZATION

Standard batch normalization (Ioffe & Szegedy, 2015) normalizes a tensor $\mathbf{X}$ along some axes, then applies a trainable affine transformation. For each slice $\boldsymbol{X}$ of $\mathbf{X}$ that is normalized independently:

$$\boldsymbol{Y} = \gamma \cdot \frac{\boldsymbol{X} - \mu_b}{\sqrt{\sigma_b^2 + \varepsilon}} + \beta$$

where $\mu_b, \sigma_b$ are mean and standard deviation statistics of a minibatch and $\gamma, \beta$ are trainable affine transformation parameters.

In our case, we do not have the memory to hold a batch of samples at a time and must compute $\mu_b, \sigma_b$ in an online fashion. To see how this works, suppose we knew the statistics of each sample $\mu_i, \sigma_i$ for $i = 1 \ldots B$ in a batch of $B$ samples. For simplicity, assume the $i^{\text{th}}$ sample is a vector $\boldsymbol{X}_{i,:} \in \mathbb{R}^n$ containing elements $X_{i,j}$. Then:

$$\mu_b = \frac{1}{B} \sum_{i=1}^{B} \mu_i \tag{23}$$

$$\sigma_b^2 = \frac{1}{B} \sum_{i=1}^{B} \frac{1}{n} \sum_{j=1}^{n} X_{i,j}^2 - \mu_b^2 = \frac{1}{B} \sum_{i=1}^{B} \left(\sigma_i^2 + \mu_i^2\right) - \mu_b^2 \neq \frac{1}{B} \sum_{i=1}^{B} \sigma_i^2 \tag{24}$$

In other words, the batch variance is not equal to the average of the sample variances. However, if we keep track of the sum-of-square values of samples $\sigma_i^2 + \mu_i^2$, then we can compute $\sigma_b^2$ as in (24). We keep track of two state variables: $\mu_s, sq_s$ which we update as $\mu_s := \mu_s + \mu_i$ and $sq_s := sq_s + \sigma_i^2 + \mu_i^2$ for each sample $i$. After $B$ samples, we divide both state variables by $B$ and apply (23, 24) to get the desired batch statistics. Unfortunately, in an online setting, all samples prior to the last one in a given batch will only see statistics generated from a portion of the batch, resulting in noisier estimates of $\mu_b, \sigma_b$.

In *streaming* batch norm, we alter the above formula slightly. Notice that in online training, only the most recently viewed sample is used for training, so there is no reason to weight different samples of a given batch equally. Therefore we can use an exponential moving average instead of a true average to track $\mu_s, sq_s$. Specifically, let:

$$\mu_s := \eta \cdot \mu_s + (1 - \eta) \cdot \mu_i$$
$$sq_s := \eta \cdot sq_s + (1 - \eta) \cdot (\sigma_i^2 + \mu_i^2)$$

If we set $\eta = 1 - 1/B$, a weighting of $1/B$ is seen on the current sample, just as in standard averages with a batch of size $B$, but now all samples receive similarly clean batch statistic estimates, not just the last few samples in a batch.

## G ONLINE DATASET

For our experiments, we construct a dataset comprising an offline training, validation, and test set, as well as an online training set. Specifically, we start with the standard MNIST dataset of LeCun et al. (1998) and split the 60k training images into partitions of size 9k, 1k, and 50k. Elastic transforms (Simard et al., 2003; Ernestus, 2016) are used to augment each of these partitions to 50k offline training samples, 10k offline validation samples, and 100k online training samples, respectively. Elastic transforms are also applied to the 10k MNIST test images to generate the offline test samples.

The source images for the 100k online training samples are randomly drawn with replacement, so there is a certain amount of data leakage in that an online algorithm may be graded on an image that has been generated from the same image a previous sample it has trained on has been generated from. This is intentional and is meant to mimic a real-life scenario where a deployed device is likely to see a restrictive and repetitive set of training samples. Our experiments include comparisons to standard SGD to show that SKS's improvement is not merely due to overfitting the source images.

From the online training set, we also generate a "distribution shift" dataset by applying unique additional augmentations to every contiguous 10k samples of the 100k online training samples. Four types of augmentations are explored. Class distribution clustering biases training samples belonging to similar classes to have similar indices. For example, the first thousand images may be primarily "0"s and "3"s, whereas the next thousand might have many "5"s. Spatial transforms rotate, scale, and shift images by random amounts. Background gradients both scale the contrast of the images and apply black-white gradients across the image. Finally, white noise is random Gaussian noise added to each pixel. Figure 5 shows some representative examples of what these augmentations look like. The augmentations are meant to mimic different external environments an edge devices might need to adapt to.

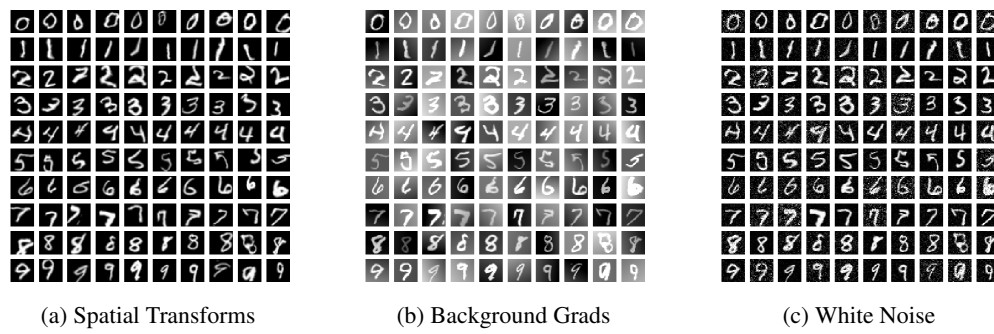

(a) Spatial Transforms        (b) Background Grads        (c) White Noise

Figure 5: Samples of different types of distribution shift augmentations.

In addition to distribution shift for testing adaptation, we also look at internal statistical shift of weights in two ways - analog and digital. For analog weight drift, we apply independent additive Gaussian noise to each weight every $d = 10$ steps with $\sigma = \sigma_0/\sqrt{1M/d}$ where $\sigma_0 = 10$ and re-clip the weights between -1 and 1. This can be interpreted as each cell having a Gaussian cumulative error with $\sigma = \sigma_0$ after 1M steps. For digital weight drift, we apply independent binary random flips to the weight matrix bits every $d$ steps with probability $p = p_0/(1M/d)$ where $p_0 = 10$. This can be interpreted as each cell flipping an average of $p_0$ times over 1M steps. Note that in real life, $\sigma_0, p_0$ depend on a host of issues such as the environmental conditions of the device (temperature, humidity, etc), as well as the rate of seeing training samples.

## H    HYPERPARAMETER SELECTION

In order to compare standard SGD with the SKS approach, we sweep the learning rates of both to optimize accuracy. In Figure 6, we compare accuracies across a range of learning rates for four different cases: SGD or SKS with or without max-norming gradients. Optimal accuracies are found when learning rate is around 0.01 for all cases. For most experiments, 8b weights, activations, and gradients, and 16b biases are used. Experiments similar to those in Section I are used to select some of the hyperparameters related to the SKS method in particular. In most experiments, rank-4 SKS with batch sizes of 10 (for convolution layers) or 100 (for fully-connected layers) are used. Additional details can be found in the supplemental code.

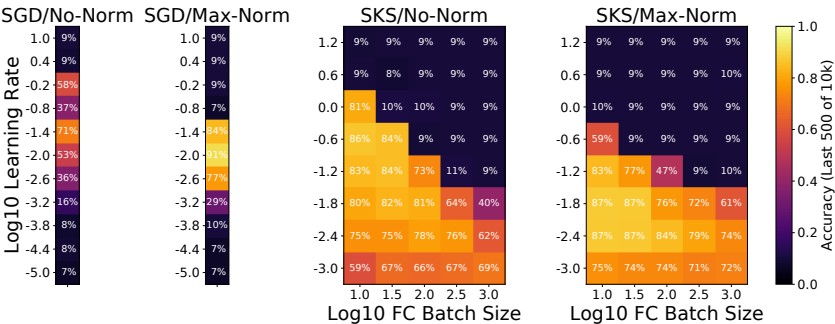

Figure 6: The left two heat maps are used to select the base / standard SGD learning rate. The right two heat maps are used to select the SKS learning rate using the optimal SGD learning rate for bias training from the previous sweeps. For the SKS sweeps, the learning rate is scaled proportional to the square-root of the batch size $B$. This results in an approximately constant optimal learning rate across batch size, especially for the max-norm case. Accuracy is reported averaged over the last 500 samples from a 10k portion of the online training set, trained from scratch.

## I    ADDITIONAL STUDIES

In Figure 7, rank and weight bitwidth is swept for SKS with gradient max-norming. As expected, training accuracy improves with both higher SKS rank and bitwidth. In dense NVM applications, higher bitwidths may be achievable, allowing for corresponding reductions in the SKS rank and therefore, reductions in the auxiliary memory requirements.

In Table 2, biased (zero-variance) and unbiased (low-variance) versions of SKS are compared. Accuracy improvements are generally seen moving from biased to unbiased SKS although the pattern differs between the no-norm and max-norm cases. In the no-norm case, a significant improvement is seen favoring unbiased SKS for fully-connected layers. In the max-norm case, the choice of biased or unbiased SKS has only a minor impact on accuracy. It might be expected that as the number of accumulated samples for a given pseduobatch increases, lower variance would be increasingly important at the expense of bias. For our network, this implies convolutions, which receive updates at every pixel of an output feature map, would preferentially have biased SKS, while the fully-connected layer would preferentially be unbiased. This hypothesis is supported by the no-norm experiments, but not by the max-norm experiments.

In Table 3, several ablations are performed on SKS with max-norm. Most notably, weight training is found to be extremely important for accuracy as bias-only training shows a $\approx 15-30\%$ accuracy hit depending on whether max-norming is used. Streaming batch norm is also found to be quite helpful, especially in the no-norm case.

Now, we explain the $\kappa_{th}$ ablation. In Section 4.1.1, we found the SVD of a small matrix $C$ and its singular values $\sigma_1, \ldots, \sigma_q$. This allows us to easily find the condition number of $C$ as $\kappa(C) = \sigma_1/\sigma_q$. We suspect high condition numbers provide relatively useless update information akin to noise, especially in the presence of $L, R$ quantization. Therefore, we prefer not to update $L, R$ on samples whose condition number exceeds threshold $\kappa_{th}$. We can avoid performing an actual SVD (saving computation) by noting that $C$ is often nearly diagonal, leading to the approximation

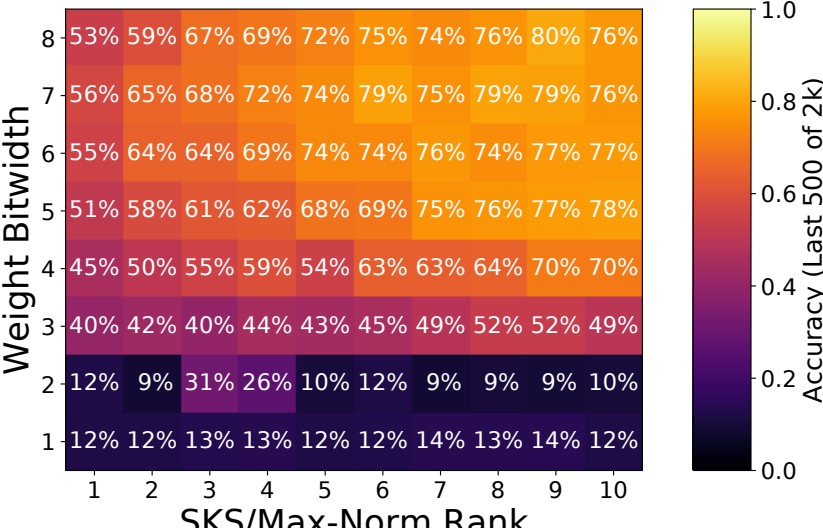

Figure 7: Accuracy across a variety of SKS ranks and weight bitwidths, showing the expected trends of increasing accuracy with rank and bitwidth. Accuracy is calculated by averaging the accuracy on the last 500 samples from a 2k portion of the training data. For bitwidths of 1 and 2, mid-rise quantization is used (e.g., 1 bit quantizes values to -0.5 and 0.5 instead of -1 and 0).

Table 2: Importance of unbiased SVD. Accuracy is calculated from the last 500 samples of 10k samples trained from scratch. Mean and unbiased standard deviation are calculated from five runs of different random seeds.

| Conv SKS | FC SKS | Accuracy (no-norm) | Accuracy (max-norm) |
|---|---|---|---|
| Biased | Biased | $79.7\% \pm 1.1\%$ | $82.7\% \pm 1.3\%$ |
| Biased | Unbiased | $83.0\% \pm 0.9\%$ | $82.4\% \pm 1.2\%$ |
| Unbiased | Biased | $77.7\% \pm 1.5\%$ | $84.6\% \pm 2.0\%$ |
| Unbiased | Unbiased | $81.0\% \pm 0.9\%$ | $83.6\% \pm 2.5\%$ |

$\kappa(\boldsymbol{C}) \approx C_{1,1}/C_{q,q}$. Empirically, this rough heuristic works well to reduce computation load while having minor impact on accuracy. In Table 3, $\kappa_{th} = 10^8$ does not appear to ubiquitously improve on the default $\kappa_{th} = 100$, despite being $\approx 2\times$ slower to compute.

Table 3: Miscellaneous selected ablations. Accuracy is calculated from the last 500 samples of 10k samples trained from scratch. Mean and unbiased standard deviation are calculated from five runs of different random seeds.

| Modified Condition | Accuracy (no-norm) | Accuracy (max-norm) |
|---|---|---|
| baseline (no modifications) | $80.2\% \pm 1.0\%$ | $83.0\% \pm 1.1\%$ |
| bias-only training | $51.8\% \pm 3.2\%$ | $68.6\% \pm 1.4\%$ |
| no streaming batch norm | $68.2\% \pm 1.9\%$ | $81.8\% \pm 1.3\%$ |
| no bias training | $81.3\% \pm 1.0\%$ | $83.0\% \pm 1.4\%$ |
| $\kappa_{\text{th}} = 10^8$ instead of 100 | $79.8\% \pm 1.4\%$ | $84.2\% \pm 1.4\%$ |

