# OpenReview forum: "Low Rank Training of Deep Neural Networks for Emerging Memory Technology"
_ICLR.cc/2020/Conference — Reject_

### Official Review · AnonReviewer2 · 2019-10-23
**Official Blind Review #2**

**Rating:** 6

**Review:**

This work presents a new online training scheme which is amenable to non volatile memories and particularly applicable to smart edge devices. This deals with 4 challenges with smart edge learning paradigms - low weight update density, weight quantization, low auxiliary memory, online learning and show experiments to understand benefits.

Disclaimer: I am far from an expert on this domain, so my review is not very well calibrated or informative.

How exactly are you implementing optLR in Section 4.2?

Maybe the section 4.3 can instead go to prior work/related work rather than be described in this paper if it’s not being used very much. It seems important for the mixing of the lower SVD elements in the OK method but perhaps can be removed.

I think for people less familiar with the area, a more intuitive explanation of section 4.4 would be helpful.

Generally how much does the algorithm suffer from using an SVD type approximation for learning?

The new contributions of this paper are to use a running estimate of Q_L, Q_R and weightings using modified Gram Schmidt.

Not immediately clear how the scheme described related to the quantization point in the challenges.

Experiments seem to show the proposed benefits but are done with artificial models/simulations. Would it be easy to implement this on chip and try it on actual hardware? Can we also compare to things like OK/UORO in the experiments?

Overall seems like a novel and interesting way to do SGD on the edge, I think that a little bit simpler explanation would make the paper more readable and some additional comparisons would also benefit the paper.

**Experience Assessment:**

I do not know much about this area.

**Review Assessment: Checking Correctness Of Derivations And Theory:**

I did not assess the derivations or theory.

**Review Assessment: Checking Correctness Of Experiments:**

I assessed the sensibility of the experiments.

**Review Assessment: Thoroughness In Paper Reading:**

I made a quick assessment of this paper.

---

> ### Author Response · Authors · 2019-11-15
> **Response**
>
> Thank you for your detailed review. We will address eight questions/suggestions/comments you raised below, in order.
> (a) optLR - We have completely reorganized the paper (see global response, point (1)). Hopefully this clears up the confusion.
> (b) Remove 4.3 - We agreed that this background was superfluous and removed it, as per global response, point (1).
> (c) Intuition for 4.4 - This is a bit difficult to do in such a short space. Our breakdown of the section into two components ((1) computation of the SVD; (2) min-var unbiased estimator of Sigma) was an attempt to provide some intuition for the result.
> (d) SVD - There is no straightforward answer to this. SVD has lower variance so in certain cases, optimization may in fact exhibit better convergence early on, as shown in the newly added convergence experiments. In deep learning experiments, however, SKS seems to have better empirical performance.
> (e) Contributions - These are not the only contributions. See global response, point (2).
> (f) Quantization - See global response, point (1). Our reorganized paper should be clearer.
> (g) Chip/UORO - See global response, point (3).
> (h) Explanation/Additional Comparisons - See global response, points (1) and (3).

---

### Official Review · AnonReviewer3 · 2019-10-23
**Official Blind Review #3**

**Rating:** 3

**Review:**

This paper proposes a low rank training method called the Streaming Kronecker Sum approximation (SKS algorithm) for training low precision models on edge devices. The authors compare their method to SGD for convolutional networks on MNIST and demonstrate improvements in terms of accuracy. The authors make use of the Optimal Kronecker-sum algorithm of Benzing et al and propose further improvements to it in the form of the SKS algorithm.

The main weakness of the paper seems to me limited experimental results - they mainly show improvements for CNNs on MNIST. The gains from their method would be made more  convincing by doing more large scale experiments on other larger datasets such as CIFAR-10/100, Imagenet and text datasets.

Overall, I recommend acceptance based on the thoroughness of the work and the authors open-sourcing their code which would additionally help with reproducibility of their results.

[Edit: Upon reading discussion among other reviewers, I have updated my score to reject.]

**Experience Assessment:**

I do not know much about this area.

**Review Assessment: Checking Correctness Of Derivations And Theory:**

I did not assess the derivations or theory.

**Review Assessment: Checking Correctness Of Experiments:**

I assessed the sensibility of the experiments.

**Review Assessment: Thoroughness In Paper Reading:**

I made a quick assessment of this paper.

---

> ### Author Response · Authors · 2019-11-15
> **Response**
>
> Thank you for your detailed review. We have addressed your concern in our global response, point (3).

---

### Official Review · AnonReviewer1 · 2019-10-26
**Official Blind Review #1**

**Rating:** 3

**Review:**

While inference on edge devices is a popular and well-studied problem in recent days, training on these devices comes with many challenges. This paper proposes a low-rank training schema that helps mitigate some of the critical challenges that occur during training models on NVM memory-based edge devices. Additionally, two techniques, namely streaming batch norm and gradient max norm, are proposed to help training in an online setting. The proposed method is mainly based on approximating the Kronecker sum and is largely inspired by (Benzing et al, ICML 2019, Optimal Kronecker-sum approximation of real time recurrent learning). The proposed approach provides a few optimizations that improves this performance further, and outperforms SGD in terms of accuracy and the number of weights updates in a limited experimental setting.

+ves:
+ Training on an edge device is a relevant setting, where there is very little work so far, and this is a useful objective.
+ Focusing on the Kronecker sum and speeding it seems like an interesting solution to solving this problem.
+ The experimental settings considered (e.g. flipping bits and adding Gaussian noise to weights) is interesting, and perhaps of larger relevance to other work in the area.
+ The ablation studies provided in the appendix are useful and appreciable.

Concerns:
- The key concern is that most of the proposed method is built upon Benzing et al’s work (ICML 2019), and the original contribution seems limited. While the paper introduces a few optimizations further, this seems to be incremental than originally novel.

- The problem is formulated for linear regression with least-squares loss, and the experimentation is carried out on the MNIST dataset (classification setting). How is the methodology relevant to the cross-entropy loss, typically used for classification? The paper does not talk about this.

- The results are shown for a network with four 3 x 3 convolution layers and two fully connected layers on the MNIST dataset. Results with different architectures on a few other datasets (at least CIFAR-10) would be necessary to assess the usefulness of this method. Some discussion on what would be the maximum depth of the network that can be trained using this training schema and hardware would be very useful.

- Do all deep neural network architectures (and loss functions) admit a Kronecker sum representation? What class of models can benefit from this method? Factorizations such as Cholesky allow to interpolate between computational complexity and decomposition rank by tuning a rank hyperparameter. Why would such factorizations not be better for memory-constrained settings?

- It would be interesting to see the results using the proposed algorithm on standard hardware, will it provide the same performance as SGD when the scale of the problem increases in terms of layers and other regularization techniques, etc.? This would help understand the performance of the algorithm in a standard-setting.

Minor issues:
- The paper could have been organized better. Most of the main paper is used to describe Benzing et al’s method, and a lot of the details of the original contributions are in the Appendix. While the equations discussed in Section 4 hold for Linear Regression, the same cannot be directly extended for other networks like CNNs. Considering all the experiments are on CNNs, Sec B.2 in the Appendix should have been in the main paper to help a reader follow the experiments.

=====POST-REBUTTAL COMMENTS========
I thank the authors for the rebuttal. The paper's organization is much better now. Unfortunately, there are many concerns which are still not convincingly answered: (i) Considering the focus of this work (as admitted by the authors in the rebuttal) is empirical and with limited technical novelty, more comprehensive results on many datasets are required. To some extent, transfer to ImageNet addresses this partially, but not convincingly enough; (ii) It is not clear from the authors' rebuttal why testing this on standard hardware is not possible/appropriate. The rebuttal says "hard to quantify due to a variety of factors" but that is vague without stating the factors.

I am willing to increase the rating from 3 to 4, but am unfortunately still towards rejecting the current shape of this work. I believe the work will benefit from considering all reviewers' comments more comprehensively to have a more impactful paper.


**Experience Assessment:**

I have read many papers in this area.

**Review Assessment: Checking Correctness Of Derivations And Theory:**

I assessed the sensibility of the derivations and theory.

**Review Assessment: Checking Correctness Of Experiments:**

I assessed the sensibility of the experiments.

**Review Assessment: Thoroughness In Paper Reading:**

I read the paper at least twice and used my best judgement in assessing the paper.

---

> ### Author Response · Authors · 2019-11-15
> **Response**
>
> Thank you for your detailed review. We will address the six concerns you raised (including one minor issue) below, in order.
> (a) We address this in the global response, point (2).
> (b) The original linear regression example was more meant to point out that the gradients of the weight matrix are sums of outer products. However, we decided that this way of organizing/motivating the paper was confusing and overly circuitous. Our new paper organization gets straight to the point in Section 3.
> (c) We address this in the global response, point (3). Also on the issue of saying anything analytically interesting, we partially address that in the global response, point (2).
> (d) Anywhere where there is a matrix-vector product should admit a Kronecker sum representation for the derivatives, and it is therefore applicable to a wide variety of deep learning models. Because Kronecker sums are the form the derivative takes, it is natural to base our low-rank representation off of this form. Keep in mind that for our application, having low-rank gradients is good, but the weights themselves do not need to be low-rank since the NVM we are using to store weights is spatially dense.
> (e) I agree this would be interesting, although hard to quantify due to the variety of factors affecting algorithm performance on any specific hardware. Our global response, point (3) may partially address this concern, since we test both SGD and SKS on a larger problem.
> (f) We address this in the global response, point (1).

---

### Official Review · AnonReviewer4 · 2019-11-01
**Official Blind Review #4**

**Rating:** 3

**Review:**

This paper proposes a low-rank training method targeting for edge devices. The main contribution is an algorithm called Streaming Kronecker-Sum Approximation. The authors claim that the proposed method addresses four key challenges of low weight update density, weight quantization, low auxiliary memory, and online learning.

The paper should be rejected because of the following reasons:
(1) The paper is a little hard to follow and the writing can be significantly improved. In particular, the authors introduce four main challenges in section 3. However, I found they are not that accessible and hard to understand. In section 4.4.2, the objective is to get a minimum variance rank-r approximation to the diagonal matrix \Sigma, but I think the authors mix "m" up with "r".
(2) The novelty of the algorithm is limited. From section 4.1 to 4.5, most discussions are about previously proposed methods. The algorithm proposed by the author (i.e., SKS) only involves some basic manipulations of linear algebra. I don't think it's novel enough to be a new algorithm.
(3) Experimental results are limited. The authors spent a lot of time discussing on-device computing, but all their experiments are just simulations on standard benchmarks. For such a paper concerning training on edge devices, I would expect to see some experiments on real edge devices.

Overall, I think the paper needs further improvements to be qualified for being accepted.

-----------------------------------------------------------------------------------------------------------------------------------------------------------------
post rebuttal:

I've read the authors' responses and the updated paper. Though my concern on writing has been resolved to some extent, I'm still unsatisfied with the empirical experiments. I believe the authors need to do experiments on edge devices since they have emphasized a lot about on-device computing. That being said, I'm not an expert in hardware and have no idea how hard it is to conduct those experiments. I've increased the score to 3 but still vote for rejection.

**Experience Assessment:**

I have read many papers in this area.

**Review Assessment: Checking Correctness Of Derivations And Theory:**

I assessed the sensibility of the derivations and theory.

**Review Assessment: Checking Correctness Of Experiments:**

I assessed the sensibility of the experiments.

**Review Assessment: Thoroughness In Paper Reading:**

I read the paper at least twice and used my best judgement in assessing the paper.

---

> ### Author Response · Authors · 2019-11-15
> **Response**
>
> Thank you for your detailed review. We have addressed your points (1-3) in the global response (1-3), respectively.
>
> One additional point you raised is that there may be a mix up of "m" with "r". m is meant to represent the singular value index (1, 2, ..., m-1, m, m+1, ..., r, q) where we should start mixing the singular values (m, m+1, ..., r, q) and therefore is distinct from r. However, you may have been referring to a symbol clash between this definition of m and the m found in the dimensions (m x n) of the weight matrix. We have relabeled some variables to avoid this confusion (m => n_i; n => n_o representing the input and output dimensions, respectively).

---

### Author Response · Authors · 2019-11-15
**Global Response to Reviewers**

We would like to thank the reviewers for their constructive criticism which has been instrumental in our paper revision process. We hope that these revisions address most of the reviewer's concerns. To reduce redundancy, we will discuss the key changes here and leave reviewer-specific comments below their respective comments.

1) Paper organization - A number of reviewers found the paper organization confusing. We have addressed this by cutting the less important background sections and being upfront (in Section 3) with the key observation that makes low rank training so well suited for NVM systems. We have also simplified this explanation significantly by focusing on two key challenges (instead of four).

2) Limited novelty - The main purpose of our paper is to make the argument that the low-rank training methods developed for RNN training applications can be used in training NVM systems. As such, it should be expected that algorithmic innovations would be more modest, in comparison to empirical tests. However, another source of novelty can come from analyzing why (or under what conditions) the technique would transfer to this new problem domain successfully. In our paper revision, we added a section analyzing convergence in the online convex setting, which can provide intuition for the key design knobs that allow effective optimization.

3) Limited experiments - Our original experiments tested performance of different algorithms across a variety of different adaptation problems. However, they were all done with the MNIST dataset. At the time, we expected this might represent a realistic use case for the first generation of test chips. However, it has since become clear that larger problems may be of interest and realistically implementable in the near future. To address these more ambitious use cases, we added a suite of transfer learning-like experiments for ImageNet. Addressing one reviewer's concerns, these experiments are swept over different algorithms, ranks, and learning rates to get a better sense of relative algorithm performance. Some reviewers also commented on the lack of on-device experiments. However, since NVM is still in its infancy, we are not currently capable of testing on physical devices. Instead, our paper aims to straddle the boundary between algorithms and hardware.

---

### Decision · Program_Chairs · 2019-12-19

**Decision:**

Reject

**Comment:**

The reviewers generally agreed that the novelty of the work was very limited. This is not necessarily a deal-breaker for a largely applied contribution, but for an applied paper, the evaluation of the actual application on edge devices is not present. So if the main contribution is the application, and there is no evaluation of this application, then it does not seem like the paper is really complete. As such, I cannot recommend it for acceptance.